# Is Low Heart Rate Variability Associated with Emotional Dysregulation, Psychopathological Dimensions, and Prefrontal Dysfunctions? An Integrative View

**DOI:** 10.3390/jpm11090872

**Published:** 2021-08-31

**Authors:** Lorena Angela Cattaneo, Anna Chiara Franquillo, Alessandro Grecucci, Laura Beccia, Vincenzo Caretti, Harold Dadomo

**Affiliations:** 1Schema Therapy Center, 21047 Saronno, Italy; dott.ssacattaneo@gmail.com (L.A.C.); stcsaronno@gmail.com (L.B.); 2Department of Human Sciences, LUMSA University, 00193 Rome, Italy; v.caretti@lumsa.it; 3Consorzio Universitario Humanitas, 00193 Rome, Italy; 4Department of Psychology and Cognitive Science, DiPSCo, University of Trento, Corso Bettini, 38068 Rovereto, Italy; alessandro.grecucci@unitn.it; 5Center for Medical Sciences, CISMed, University of Trento, 38122 Trento, Italy; 6Neuroscience Unit, Department of Medicine and Surgery, University of Parma, 43125 Parma, Italy; harold.dadomo@gmail.com

**Keywords:** heart rate variability, polyvagal theory, neurovisceral integration model, emotional regulation, psychopathology, prefrontal functions

## Abstract

Several studies have suggested a correlation between heart rate variability (HRV), emotion regulation (ER), psychopathological conditions, and cognitive functions in the past two decades. Specifically, recent data seem to support the hypothesis that low-frequency heart rate variability (LF-HRV), an index of sympathetic cardiac control, correlates with worse executive performances, worse ER, and specific psychopathological dimensions. The present work aims to review the previous findings on these topics and integrate them from two main cornerstones of this perspective: Porges’ Polyvagal Theory and Thayer and Lane’s Neurovisceral Integration Model, which are necessary to understand these associations better. For this reason, based on these two approaches, we point out that low HRV is associated with emotional dysregulation, worse cognitive performance, and transversal psychopathological conditions. We report studies that underline the importance of considering the heart-brain relation in order to shed light on the necessity to implement psychophysiology into a broader perspective on emotions, mental health, and good cognitive functioning. This integration is beneficial not only as a theoretical ground from which to start for further research studies but as a starting point for new theoretical perspectives useful in clinical practice.

## 1. Introduction

The sympathetic or parasympathetic reactivity of the autonomic nervous system (ANS) has often been cited as one of the most critical factors influencing susceptibility to stress due to its crucial role in mobilizing biological resources during acute “fight or flight” responses to threatening environmental events [1,2]. Individuals often show vast differences in autonomic reactivity, which has been associated with a variety of disorders and pathological conditions, from chronic stress-related disorders to psychopathology [3,4]. Although many authors have identified the relationship between the ANS, psychological functioning, and psychopathology, a comprehensive model of how these factors interact is still lacking. Studies that try to connect the heart and the brain networks via the vagus nerve can serve as a support for this understanding. The vagus nerve supports the communication between the heart and the brain, especially during emotional reactions, and its contribution has been known for a hundred years. In the 19th century, Bernard [5], who gave a significant contribution to modern physiology, stated the concept of milieu intérieur (translated as the “internal environment”). He stated that “The fixity of the milieu supposes a perfection of the organism such that the external variations are at each instant compensated for and equilibrated. All of the vital mechanisms, however varied they may be, always have one goal, to maintain the uniformity of the conditions of life in the internal environment. The stability of the internal environment is the condition for the free and independent life” [5].

Bernard’s conclusions were drawn by observing that the heart’s affections rebound on the brain activity and vice versa through the vagus nerve. This is along with previous [6] and recent [7] considerations on the existence of specific multilevel control processes from the brain to the different organs, creating a bidirectional connection between the heart and the brain. Porges [8] introduced the Polyvagal Theory, explaining the role of the vagus nerve as a regulator of the internal viscera and as a mediator of the communication between the heart and the brain. This theory stresses the importance of autonomic functions in regulating human behavior in normal and abnormal conditions. The vagus nerve originates into two different nuclei of the brainstem: the Dorsal Motor Nucleus of the vagus (DNMX) and the Nucleus Ambiguus (NA), each of them ending in the sinoatrial node, but only NA having the control of respiratory sinus arrhythmia (RSA) which is an index of cardiac vagal modulation, and it is associated with emotion regulation (ER) [9].

The DNMX and NA act competitively on the sinoatrial node, adjusting the anabolic parasympathetic activity and the catabolic sympathetic one. In this way, these two branches of the vagus’s independent action have different effects on RSA and HRV. The sympathetic nervous system (SNS) innervates the cardioaccelerating center of the heart, the lungs (increased ventilatory rhythm and dilatation of the bronchi), and the non-striated muscles (artery contraction), releasing adrenaline and noradrenaline. On the contrary, the parasympathetic nervous system (PNS), which uses the neurotransmitter acetylcholine (ACh), innervates the cardiomoderator center of the heart, the lungs (slower ventilatory rhythm and contraction of the bronchi), and the non-striated muscles (artery dilatation), reducing the experience of stress. These two systems act agonistically on the heart, respectively, through the stellate ganglion (a collection of sympathetic nerves) and the vagus nerve (a parasympathetic nerve). The interaction of these two branches on the sinoatrial node originates the cardiac variability, measured using electrocardiography by monitoring heart rate variability (HRV), which is a beat-to-beat variability [10]. HRV can be considered as an indicator of physiological stress or arousal. The frequency-domain analysis typically includes three measures: very low frequency (≤0.04 Hz), low frequency (LF, 0.04–0.15 Hz), and high frequency (HF, 0.15–0.4 Hz). The HF component measures vagal activity, while the LF component is related to a combination of both vagal and sympathetic activities, and LF/HF ratio reflects the cardiac sympathovagal balance [11].

Porges [8] adopts a phylogenetic perspective and proposes that mammals, but not reptiles, have a brainstem organization characterized by a ventral vagal complex (including NA) that influences attention, emotion, motion, and communication. He also suggests, with his theory, explanations on how heart rate changes with novel environmental stimuli. Indeed, according to the Polyvagal Theory, there are three evolutionary phylogenetic stages behind the development of the vagus nerve. The dorsal vagal system (archaic, unmyelinated) is phylogenetically the oldest one, and it is associated with immobilization (death feigning, vasovagal syncope, and behavioral shutdown). The sympathetic vagal system is associated with the fight or flight response (active avoidance of the threat). Finally, the last step of this evolution recognizes the ventral vagal system (newest and myelinated), which is associated with social communication and behaviors (facial expression, vocalization, listening).

This model must be considered hierarchical. The last system, named the myelinated vagal system, is the first to be engaged in social and complex human experience. When this system fails in its functionality, the sympathetic vagal system is engaged by displaying the fight or flight response behavior. If this structure fails, the most ancient structure is engaged with the immobilization response [8,12]. In this pattern, psychopathology arises when the latest structures fail in their functionality, implying a significant dysfunction in ER.

Beauchaine and Thayer [13] stress the validity of respiratory sinus arrhythmia (RSA) as a transdiagnostic biomarker of emotional dysregulation and concomitant psychopathology. Basing his assumption on the RDoC (Research Domain Criteria) project for the re-conceptualization of psychopathology [14,15,16,17], they underline the importance of psychophysiological measures as useful tools to use in order to address core psychopathological transdiagnostic mechanisms.

The vagus nerve originating in the DNMX is associated with reflexive regulation of visceral functions, while the vagus nerve originating in the NA is associated with active processes like attention, motion, emotion, and communication. According to Porges [18], mammalians reach the homeostatic balance through bidirectional communications between the peripheral organs and the brain. This entire process is supported by neuroception, a term coined by Porges to describe the mechanism by which our brain can detect dangerous environmental stimuli by analyzing the information coming from our senses through body scanning. When our brain detects (consciously or unconsciously) a threat, disruption of homeostasis happens. Neuroception determines the connection between environmental aspects and specific physiological states that support either fight-flight or social engagement behaviors. The detection of the possible threat activates our organism, leading it to a stress state, which is to say in a disorganization of the autonomic system’s rhythmic structure and, consequently, behavioral one if an invalid neuroception of safety and danger occurs. If there is no coherence between the detection of the risk and the visceral response to risk, a dysfunctional, maladaptive physiological reactivity may happen in the long term” [18].

We can estimate the stress degree-level of an organism by measuring RSA, estimated by heart rate variability (HRV) that registers increased heart rate during inspiration and a decrease during expiration. The Polyvagal Theory [19] also provides a plausible explanation for the correlation between atypical autonomic regulation (e.g., reduced vagal influence on the heart) and psychiatric and behavioral disorders, outlining a complex framework of human thinking and behavior.

In a similar vein, Thayer and Lane [20,21] introduced the neurovisceral integrated model of the heart-brain activity, in which prefrontal and limbic structures control HRV. They describe a model in which HRV is both related to attentional and affect regulation. With this model, they deepened the understanding of the central autonomic network (CAN), already pointed out by Benarroch [22], and concluded that it is an integrated component of a complex regulation system by which the brain controls visceromotor, neuroendocrine, and behavioral responses, which are essential for goal-directed behavior and human adaptability too. The CAN includes several regions of the central nervous system (CNS) such as the anterior cingulated, insular, orbitofrontal, and ventromedial prefrontal cortices together with the central nucleus of the amygdala (CeA), the paraventricular and related nuclei of the hypothalamus, the periaqueductal gray matter, the parabrachial nucleus, the nucleus of the solitary tract (NTS), the nucleus ambiguus (NA), the ventrolateral and ventromedial medulla, and the medullary tegmented field. All of these regions are reciprocally interconnected so that information can flow bidirectionally between lower and higher brain levels.

It is evident that all these regions intervene in modulating human behavior by connecting executive functions (prefrontal cortices), physiological reactions, and the autonomic response through the NA and the vagus nerve activities that regulate the sinoatrial node [8]. Thayer et al. [23] pointed out a link between stress, HRV, and cognitive deficits, hypothesizing that HRV indexes important aspects of prefrontal neural function. These assumptions come upon neuroimaging evidence that asserts that the primary output of the CAN is mediated through preganglionic sympathetic and parasympathetic neurons, which innervate the heart via the stellate ganglia and the vagus nerve, respectively. After all, there is increasing evidence in the literature that high HRV correlates with better neuropsychological performances (especially working memory, attentional set-shifting, and response inhibition), despite low HRV that correlates with worse neuropsychological performances [23,24,25,26,27,28] In their review, Thayer and Lane [21] also indicate low vagally mediated HRV as an endophenotype for a range of physical and psychological disorders, including psychopathology.

Our paper reviews the scientific works published on the relation between HRV and emotions, HRV and psychopathology and HRV and neuropsychological functions. We aim to define the state-of-art on this topic to delineate new research projects in the field even if no such comprehensive work exists to the very best of our knowledge.

We hypothesize that HRV is a variable that influences different dimensions such as executive functions, ER, and psychopathology, and we decided to deepen these three variables from this hypothesis because they represent essential elements in influencing psychological health and different psychotherapeutic constructs (See Figure 1).

Therefore, this review intends to evaluate and emphasize the relationship between HRV emotion regulation, executive functions, and psychopathology and how these can be conceptualized according to the Polyvagal theory.

## 2. Emotional Dysegulation and Heart Rate Variability

Evidence shed light on the link between HRV and emotional responses [8,12,19,29,30,31]. Participants with higher baseline HRV exhibit appropriate emotional responses during fear-potentiated startle responses and phasic heart rate responses [27,32,33]. By contrast, participants with low baseline HRV are slower in recovering from psychological stressors of cardiovascular and immune responses than controls. This evidence confirms that HRV is an index of self-regulation and consequent subjective well-being [34]. Porges [29] has been one of the first to underline the link between high HRV with adaptive emotional regulation (ER) and coping strategies and between low HRV with emotional dysregulation visible in behaviors characterized by anxiety and rigid attentional threat processing. Emotion regulation ability is associated with greater baseline HRV [21,30] and task-related HRV during successful ER [35,36].

Notably, phasic increases in HRV in response to emotion-inducing situations are associated with better and effective ER [37]. One study showed that the use of reappraisal or suppression strategies during successful ER is associated with increased HRV [38]. From a neural point of view, an increase in HRV during successful emotion regulation is associated with cerebral blood flow changes in areas relevant for ER and inhibitory processes [39].

Still, Thayer and Lane [20] consider cardiac vagal tone indexed by HRV to measure both the integrity and functionality of neural networks involved in emotion-cognition interactions. High HRV portrays the integrity and the healthy functions of these neural networks compared to low HRV, representing a disintegration of their functionality. According to McCraty and Shaffer [40], there is a link between higher levels of resting vagally-mediated HRV and the performance of executive functions such as attentional and emotional processing by the prefrontal cortex. Moreover, the authors report that HRV can be seen as an index of resiliency and flexibility because it indicates the ability to self-regulate and adapt to challenging situations and threatening events.

Appelhans and Luecken [30] claim that HRV is an index of emotionality and considered pain as a homeostatic emotion with an inverse association between LF and pain sensitivity (Appelhans and Luecken) [41]. According to the authors, pain is a homeostatic emotion that is influenced by the affective system in its different parts and global characteristics, supporting, with this perspective, the model of neurovisceral integration [20,27], which sees the CAN as responsible for some aspects of homeostatic regulation. According to Park and Thayer [31], HRV associates with top-down and bottom-up cognitive processes of emotional stimuli, and higher resting HRV is linked with more functional and efficient top-down and bottom-up cognitive processing of emotional stimuli and a consequent more efficient ER. In contrast, lower resting HRV is linked with hypervigilant and maladaptive cognitive responses to emotional stimuli, representing an obstacle to adaptive ER. They also suggest that maladaptive cognitive processes of emotional stimuli (observed in people with low HRV) may contribute to health issues observed in a wide range of people with low HRV.

The Model of Neurovisceral Integration suggests that vagally mediated heart rate variability (vmHRV) represents a psychophysiological index of inhibitory control and is associated with emotion regulation capacity. A study from Visted et al. [42] explored the correlation between ER abilities assessed using the Difficulties in Emotion Regulation Scale (DERS) [43]. They found that difficulties in ER negatively correlated with resting vmHRV, with specific troubles linked to the inability to behave following personal goals. This evidence confirms what was previously found by Williams et al. [44] that using the DERS scale reported a significant negative association between resting vmHRV and difficulties in ER, with problems linked to anxiety and ruminative tendencies.

The link between HRV and cognitive functions does not confine only to top-down inhibitory processes but extends to other cognitive domains.

In an interesting work, Xiu et al. [45] connected working memory, HRV, and ER, suggesting that working memory training could improve ER abilities. Specifically, they found that high frequency-heart rate variability (HF-HRV) increased after 20 days of working memory training in the ER condition, meaning that working memory training can influence ER. Even though there is a need for more studies to confirm these results, they represent a relevant indication of the correlation between cognition-emotion and evidence that higher resting HRV is mainly associated with flexible and adaptive top-down and bottom-up cognitive processing. These adaptive cognitive skills contribute to effective ER. In contrast, lower resting HRV seems to associate with hypervigilant and maladaptive bottom-up and top-down cognitive responses to emotional stimuli, making this cognitive deficiency deleterious for ER and confirming what was previously said. For this reason, we can claim that maladaptive cognitive processing of emotional stimuli observed in people with lower HRV may be disadvantageous for emotional and physical health, and this could explain why low HRV occurs in people within a wide range of psychopathologies.

Beauchaine and Thayer [13] state that emotional dysregulation is related to poor executive control over behavior because of the structural and functional connections between the prefrontal cortex and the parasympathetic nervous system via the vagus nerve (complex interactions between cortical and subcortical pathways, including amygdala circuits). In the regulation and modulation of negative emotional stimuli, such as anxiety which create many cognitive, somatic, and behavioral responses, among the limbic structures, the amygdala appears to have an essential role in regulating emotions. Anxiety-related responses are modulated by GABAergic modulation, and neurosteroids seem to modify, interacting with GABA, neuronal excitability. This evidence makes neurosteroids representing a core for developing new anxiolytic drugs [46] Furthermore, anxiolytic drugs seem to have good effects on amygdala functionality that link with anxiety disorders when dysregulated [47]. Less self-reported anxiety seems to be associated with diminished activity in areas connected with negative emotions and increased activity in regions linked to regulatory processes after administering allopregnanolone, a progesterone-derived neurosteroid known for its anxiolytic properties [48].

Given the importance of HRV on a whole series of parameters and variables, it has been shown how antiarrhythmic agents alter HRV with a direct effect on the ANS and myocardial contractility. Different comparative analyses showed that Amiodarone, despite interacting with the ANS centrally [49] and peripherally [50,51] did not affect HRV. On the other hand, Flecainide and Propafenone have vagolytic [52] and beta-blocking [53] properties, which could further modulate sympathetic and parasympathetic activity in the heart [54,55] decreasing all the parameters of HRV in the time domain and the frequency domain including the markers of vagal activity [56]. On the contrary, Oymatrine, compared to Propafenone, increases HRV [57]. Although this may prove to be an essential field of study, caution in using these drugs to manage HRV is imperative. It is well known how much some potent antiarrhythmic drugs may increase the incidence of sudden death, as observed by different studies [58].

## 3. The Importance of Emotional Regulation and Heart Rate Variability in General Psychopathology

As seen above, high HRV is associated with a successful adaptation [12,19,23,31]. Adapting to different environmental stressors determines good individual functionality, but subjects with varying degrees of psychopathology seem to lack this capacity. As said before, since ER is an essential skill for psychological health and it represents one’s ongoing adjustment to continuous environmental stimuli and changes [59], an adequate emotional ability is crucial for general health since it facilitates the selection of optimal responses by inhibiting and rejecting dysfunctional options [21]. By contrast, links between low HRV and psychopathology are emerging. A meta-analysis by Zahn et al. [60] supports the notion of a relationship between low HRV and worse self-control in inhibiting or diverting dominant impulses related to dysfunctional thoughts, behaviors, and emotions [61].

Regarding anxiety disorders, low HRV can be considered an endophenotype of panic disorder [62,63]. A study from Zhang et al. [64] explored LF/HF in patients with panic disorders, knowing that notably, LF/HF ratio seems associated with sympathetic modulation. Patients with PD exhibited an impairment in sympathovagal modulation compared to healthy controls, corroborating the idea that an autonomic imbalance in patients with PD is the consequence of mental stress, which causes this autonomic imbalance. HRV is heritable [65,66,67] and is state-independent. Namely, it also occurs in the absence of panic symptoms [68,69], co-aggregates within family members [62,63], and is lower in children of patients with panic disorder than in children of healthy controls [70]. All of these observations confirm the fact that low HRV is an endophenotype for panic disorder.

An interesting association also exists between reduced HRV and epilepsy [71]. The autonomic imbalance in patients with epilepsy can also represent a risk for cardiovascular disease, and this means that HRV can be used as a guide to prevent and assess patients with risk for cardiovascular disease. Some recent evidence shows correlations between low HRV and schizophrenia [72,73]. A literature review from Guccione et al. [74] highlights that several studies had demonstrated a sympathovagal imbalance in individuals diagnosed with schizophrenia. A study from Castro and colleagues [75] demonstrated that schizophrenic patients showed difficulties in recovering HRV.

Meyer et al. [76] showed that individuals diagnosed with borderline personality disorder (BPD) and with post traumatic stress disorder (PTSD) exhibit lower root mean square of the successive differences (RMSSD) compared to healthy individuals. RMSSD is another index used to measure HRV. Impairment in RMSSD may have a connection with early maladaptive experiences and traumatic events. Interestingly, Dixon-Gordon et al. [77] proved that individuals with BPD after using acceptance strategy exhibited high HRV, indicating that such populations can benefit from ER training.

The inability to regulate emotions and behaviors is also typical of attention deficit hyperactivity disorder (ADHD). Rukmani et al. [78] screened out 270 gender AHDH children (7–12 years) for their investigation and selected 10 children without psychiatric and neurological comorbidities. They found that ADHD children presented a sympathovagal imbalance, characterized by a reduction in overall HRV with a sympathetic predominance.

Neuroticism, a personality trait characterized by negative affect, higher anxiety, and major reactivity to external stress, seems to predispose individuals for psychopathology such as schizophrenia [79] and to link with more difficulties in regulating negative emotions. In this regard, according to Di Simplicio et al. [80], during a negative emotion challenge, individuals with high neuroticism traits reported reduced high-frequency HRV (HF-HRV). Since this index represents the parasympathetic part of the systems showing the flexibility of the vagal tone, these results show that people with high negative affectivity (anxiety, depression) and high reactivity have difficulties modulating their physiological response.

A recent meta-analysis from Koch et al. [81] showed that individuals diagnosed with major depression (MD) exhibited a significant reduction in HF-HRV, LF-HRV, LF-HF ration, RMSSD, and, in general, in all HRV measures compared to controls. Although it is widely recognized that a high value in HRV is related to psychophysical well-being, some authors have also found an elevation of HRV in anorexic patients. In this regard, it may be important to evaluate the hypothesis that there may be an ideal HRV range. In fact, a recent meta-analysis evaluated this hypothesis and discovered a distinct U-shaped pattern, with healthy controls clustered towards the center, individuals with anorexia nervosa experienced increased HRV, and all other disorders were associated with lower HRV parameters. This metanalysis has the advantage of opening a crucial and original question, so it would be helpful to verify the assumption experimentally despite some limitations. [82]. In general, all these results strengthen the idea that HRV can be considered a transdiagnostic index for stress, consequent cardiovascular diseases, and generally worse health outcomes.

## 4. Heart Rate Variability and Neuropsychological Functions

Thayer et al. [23] pointed out the existence of a relationship between HRV and prefrontal neural functions [39,83,84,85]. Some findings suggest that cortical activity tonically inhibits brainstem cardioacceleratory circuits and corroborates an association between HRV and the medial prefrontal cortex activity. Lane et al. [39] studied the correlation between a spectrally derived index of vagally mediated HRV, the high frequency-HRV (HF-HRV), and cerebral blood flow data obtained by PET. In this study, HF-HRV correlates with blood flow in the right superior prefrontal cortex (BA 8, 9), with the left rostral anterior cingulate cortex (BA 24, 32), with the right dorsolateral prefrontal cortex (BA 46), and the right parietal cortex (BA40). At the same time, emotional arousal shows an association with a decrease in HRV and a concomitant decrease in the activity in the same regions, indicating that high HRV correlates with prefrontal activation both during emotional and neutral situations.

In contrast, low HRV correlates with lower cerebral activity in the same region only during emotional arousal. These outcomes confirm that HF-HRV links to better cognitive performances in threat and non-threat conditions, while LF-HRV correlates to improved cognitive performances only in threat conditions. In line with these results, we can deduce that low HRV seems linked only with emotional arousal and worse prefrontal performances than high HRV. These findings agree with assumptions from Ter Horst [86] about a general inhibitory role of the prefrontal cortex on heart activity via the vagus nerve. All of these results give strong evidence of the critical role of the prefrontal cortex in the modulation of subcortical cardioacceleratory circuits via an inhibitory pathway associated with vagal function. These findings make consistent the assumption that HRV can be an index of neuropsychological functions, such as attentional, set-shifting, and planning abilities. Besides, further studies suggest that the right prefrontal cortex is preferentially related to inhibitory processes across the different cognitive, motor, and affective tasks [87,88,89,90]. Therefore, we can deduce that the right hemisphere may be preferably involved in inhibitory processes, useful for cognitive, affective, and physiological regulation [23,27]. Such alterations in the prefrontal cortex have several implications for psychopathology. Dysfunction in the prefrontal cortex (also named prefrontal dysfunction, or prefrontal executive dysfunction) is characterized by functional (blood perfusion) or structural (grey and white matter) alterations. It has been clinically proven to result in impulsivity, compulsivity, risk taking, impaired self-monitoring, difficulty in disengaging from ruminative thoughts, enhanced stress reactivity and lack of top-down regulation of emotional responses [91]. According to a recent perspective, prefrontal dysfunction characterizes addictions, depression, schizophrenia, and personality disorders [92].

Moreover, inhibitory processes are the core dimension of several neuropsychological functions involving working memory, such as active short-term storage, online processing, and manipulation of information [93]. This working memory is indicated, by broad literature, as the nucleus of prefrontal functioning, namely attentional processes [94,95,96]. The hypothesis that HRV has reliability in indexing prefrontal activity [23] comes from the study of Hansen et al. [24], where they measured HF-HRV and LF-HRV of the military personnel while performing attentional and memory tasks and non-executive tasks (simple reaction time and response latencies to specific stimuli). The study’s outcome showed better cognitive performances in the HF-HRV group than the LF-HRV group (faster reaction times and fewer false-positive responses). More in detail, the HF-HRV group performed better both in executive and non-executive tasks, while the LF-HRV group performed worse only in the executive tasks but not in the non-executive ones. Hence, it seems that HRV is connected only with executive tasks and does not differentiate non-executive performances. A second scientific work from the same author replicated these results [26], and in this study, they looked at the correlation between HRV and cognitive functions in threat and non-threat conditions, analyzing 65 male sailors (mean age 23.1) from the Royal Norwegian Naval Academy. While recording participants’ HRV, they displayed them a computerized version of two cognitive tests: the Continuous Performance Test (CPT) [97] in its California Computerized Assessment Package abbreviated version (CalCAP) used for assessing sustained attention, and four sub-tests such as Simple Reaction time Task (SRT), Choice Reaction Time Task (SRT), Serial Pattern Matching 1 (SPM 1), and a Serial Pattern Matching 2 (SPM 2); and a modified version of the Working Memory Task (WMT) from Hugdahl et al. [98] with a task which was an n-back task (2-back task). The sample was divided into threat and non-threat subsamples, and it was administered an electrical shock (unpleasant but not painful) by a pulsating (18Hz) adjustable DC shock generator in the second group but not in the first one.

The results found out that there are individual differences in autonomic, cognitive, and behavioral aspects of emotional regulation both in ordinary and challenging contexts. In fact, in scenarios requiring vigilance, high HRV subjects demonstrated a better capacity to hold prolonged focused attention than low HRV subjects. Moreover, subjects with low HRV appeared more sensitive to environmental changes than high HRV ones. Furthermore, high HRV subjects showed superior cognitive performance in threat and non-threat conditions, while low HRV subjects showed bad performances during non-threat conditions and improved performance in threat conditions. These outcomes seem to delineate that poor neuropsychological performances can be found in subjects with low HRV at rest. These data confirm previous hypotheses from Frankenhaeuser et al. [99] and Broadbent [100] about individual differences in cognitive performances due to specific physiological patterns and specific environmental stimuli. Frankenhauser et al. [99] prior observed that subjects with high HR and low HR performed better respectively in a contest of understimulation and overstimulation. Stenfors et al. [101] measured executive functions and cardiovascular parameters, analyzing HRV indices in 119 healthy working adults (79% female) and focusing on Standard Deviation of NN (SDNN), Root of the Mean Squares of Successive Differences (RMSSD), High Frequency (HF) power band from spectral analysis, and QT Variability Index (QTVI). They also included specific adjustments for demographic variables such as age. The outcomes show that age seems responsible for the confusion in the correlation between HRV and executive functions and explains the association between executive measures with SDNN and RMSSD parameters. However, there is an index called QTVI that the age variable does not invalidate. This last parameter proves a clear correlation with prefrontal performances: indeed, while low QTVI registers better prefrontal performances, high QTVI correlates with worse prefrontal performances, specifically for inhibition, shifting, updating, and speed capacities.

In contrast, no correlations were observed between any cardiovascular parameter and working memory task performances. These data increase our understanding of how external variables (e.g., age, education degree, level of physical activity) can affect the correlation between HRV parameters and performances in executive functions, suggesting that, because of its independence from age, QTVI is an index better should be used. Because of that, future studies are needed to deepen the clarification of external variables’ potential interference in the correlations between heart indices and brain performances.

Moving on, Gathright et al. [102] investigated executive functions’ hypothetical role in mediating depressive symptoms, measured by BDI-II, the Beck’s Depression Inventory [103], and resting HRV in heart failure patients. Analyzing 109 patients with HF (Heart Failure), the authors found an association between higher BDI-II scores and lower resting HF-HRV among participants with poorer executive functions. This evidence can suggest two interpretative hypotheses: in HF-HRV patients, there are similar structural brain changes responsible for lower executive functions, increased depression, and poorer autonomic functioning, whereas individuals with good executive functions keep a healthy lifestyle that does not allow depression to impact negatively on the autonomic function. Evidence of the link between HRV and prefrontal functions comes from the study of executive functions in different disorders. Recent studies point toward the direction of structural [104] and functional [105] abnormalities in psychopathy when performing executive tasks [106] Morgan and Lilienfeld [104], in their meta-analytic review of thirty-nine studies about the relationship between psychopathy and executive functions, found that antisocial groups performed 0.62 standard deviations worse on executive function tests in comparison to control groups. Gorenstein [105] has found several neuropsychological tests measuring prefrontal functions (namely Wisconsin Card Sorting Task, Sequential Matching Memory Task, and Necker Cube Task), psychopaths exhibit the same patterns of frontal lesion patients. Based on such assumption, he concluded that psychopathy relies on deficits associated with the frontal lobe dysfunction, but in contrast, Hare argued that Gorestein’s conclusions are undermined by inhomogeneous samples regarding age, education, IQ, and substance use [90]. He replicated Gorestein’s experiment with forty-six convicts and carried out a series of variance and covariance analyses using exact age, education, IQ, and substance use as covariates, not finding any group differences in task performance.

Still, Hansen et al. [107] attempted to study the link among HRV, psychopathy dimensions, and neuropsychological function, using Hare’s four-facet model, the continuous performance test (CPT), and a working memory test to study the relationship between all these variables.

Before describing the evidence determined from this study, it is needed a description of Hare’s four-facet model, which provides a representation of psychopathy in four facets: interpersonal style (the tendency to manipulate other subjects, to act pathological lying, and to expose a grandiose sense of self-worth), affective style (characterized by lack of empathy and remorse or guilt), impulsive lifestyle (typified by sensation-seeking and irresponsibility), antisocial behavior (defined by the use of violence). They examined 33 male prisoners and found that the interpersonal style facet showed a positive relationship with HRV during baseline. The interpersonal style facet showed the most substantial influence on HRV during the test conditions and exhibited better performance than those with low scores on cognitive tasks involving executive function. This evidence suggests that psychopathy might have different underlying physiological and cognitive mechanisms and that HRV seems associated with specific psychopathy facets and specific cognitive mechanisms. Moreover, they seem to give evidence of an association among low HRV, worse cognitive prefrontal performances, and several psychopathological dimensions but primarily for the inhibitory one. The autonomic imbalance, considered an index of disinhibition of sympathoexcitatory neural circuits usually under tonic inhibitory control via the prefrontal cortex, could be the final common pathway linking psychosomatics psychopathology [27].

In conclusion, at this stage of the debate, it is clear that there is a correlation between heart activity and prefrontal brain activity. Actual scientific data suggest that the prefrontal cerebral regions may play a part in influencing heart activity, especially for their inhibitory role mediated by the vagus nerve activity on the sinoatrial node. According to the above evidence, research needs further studies are required to clearly understand which heart activity parameters are strictly connected with specific prefrontal cerebral tasks.

## 5. Conclusions

HRV has traditionally been treated as a simple, one-way, dependent variable to be observed to assess the influence of heart rate on global sympathetic and parasympathetic regulatory systems [18]. However, the perspective that emerges from this research shifts our attention to a complex system that incorporates and influences complex neurophysiological mechanisms, adaptive functions, and above all, it is a bidirectional system between central elements and peripheral/autonomic elements.

In line with this, there is evidence in the literature of a strict relation between HRV, executive functions, and emotional dysregulation. In particular, there is evidence of a solid correlation between high resting HRV and better cognitive functions, especially executive functions [23,39,83,84,85]. Subjects with high rest HRV can dispose of a better skill in adaptation to environmental stressors and better cognitive responses to emotional stimuli. This means being able to process and react in a functional way to emotional stimuli and distress.

Conversely, people with low HRV show worse activation in the prefrontal cortex, the rostral cingulate cortex, and the parietal cortex [39], and worse ability to dominate mental and behavioral impulses [48]. Furthermore, there is evidence of a correlation between low HRV and dysfunctional ER [29,39,42,108].

Being that emotional dysregulation the basis of many psychopathological dimensions, we can state that HRV is linked to psychopathology (See Figure 2).

Evidence of correlation between low HRV and anxiety [63], panic disorder [64], epilepsy [71], schizophrenia [72,73], personality disorders [76], and ADHD [78] are given.

There are also suggestions that low HRV could be an endophenotype of specific psychopathologies (e.g., panic disorder), providing data from neurophysiological-cardiovascular level to interpret and test hypotheses relating to psychological processes such as ER and cognitive performances.

In fact, as conceptualized by the Polyvagal Theory, physiological states may influence a wide range of social behaviors emitted, such as the ability to regulate emotional expressions and neural regulation of the social engagement system. This framework explains how HRV may be a marker of specific central pathways activation coming from cortical and subcortical areas (involving the temporal cortex, the central nucleus of the amygdala, and the periaqueductal gray) involving the regulation of both the vagal component and the somatomotor component of the social engagement system.

This perspective may drive research towards new specific hypotheses or neural mechanisms and mediators, opening fundamental questions about the adaptive characteristics of specific psychophysiological responses. More studies are needed to confirm this hypothesis and determine if low HRV can be considered a predictor of psychopathology mediated through cognitive dysfunctions. It is explicit that a relation between heart and brain exists and good heart functionality with mental, emotional, and physical health. Future research is needed to deepen which HRV parameters are linked with specific neuropsychological functions that may undergo bad cognitive performances, ER, and consequent psychopathological dimensions.

This evidence would provide the possibility to integrate the assessment of psychophysiology into the comprehension of psychopathological features and cognitive issues, providing future directions for improvement in research and on the assessment in clinical practice. In fact, given the role of the ANS flexibility and adaptability, future research may account for autonomic problems, even in psychopathological conditions considered at-risk in order to improve the evaluation of situations acknowledged prodromic for worse outcomes.

## Figures and Tables

**Figure 1 jpm-11-00872-f001:**
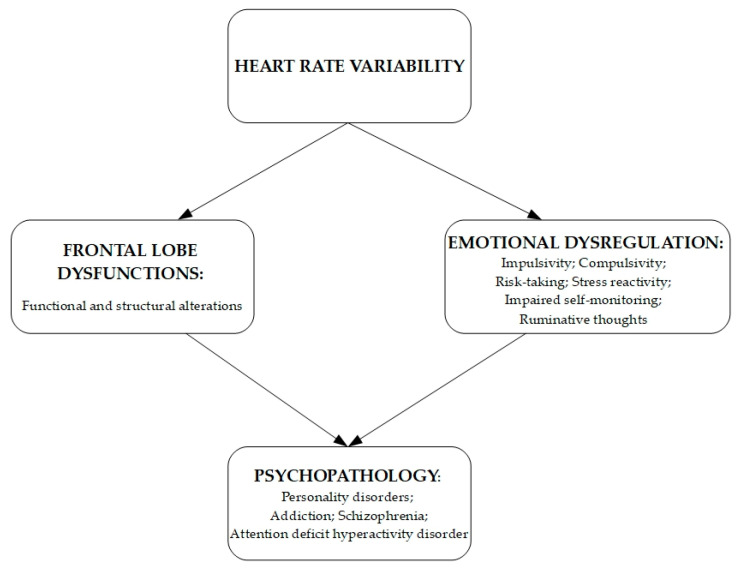
HRV is a variable that in influencing different dimensions such as Frotal Lobe Functions and Emotion Dysregulation contributes to the development of psychopathology.

**Figure 2 jpm-11-00872-f002:**
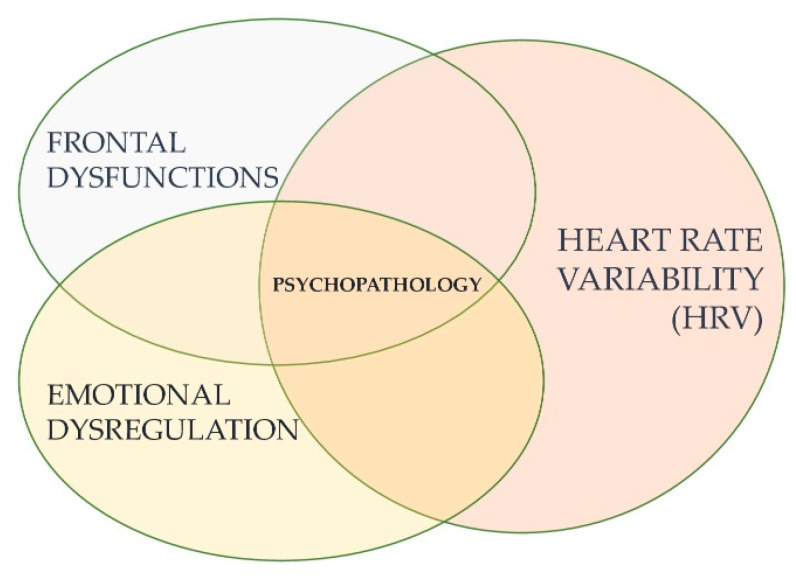
The relation between low HRV, executive functions, and emotional dysregulation is linked to psychopathology.

## Data Availability

This study did not report any data.

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
