# Peer review of "Is Low Heart Rate Variability Associated with Emotional Dysregulation, Psychopathological Dimensions, and Prefrontal Dysfunctions? An Integrative View"

_jpm, 2021, doi:10.3390/jpm11090872_

Round 1

Reviewer 1 Report

This paper discusses the relation between the heart rate variability(HRV) and emotional regulation, neuropsychological functions and psychopathology.  Summarizing these findings will lead to the next psychosomatic correlation study. 

Minor comments:

  1. Isn't it necessary to specify the criteria for the cited paper? For example, like this review paper, Heart rate variability as a biobehavioral marker of diverse psychopathologies: A review and argument for an "ideal range".  Heiss,S, Vaschillo,B al., Neurosci Biobehav Rev. 2021 Feb;121:144-155.  doi: 10.1016/j.neubiorev.2020.12.004. Epub 2020 Dec 9.

It is better to show that the authors are not arbitrarily collecting papers.  It is even better to state if there are any disproving papers.

  1. L89

Although author cited 11th paper for explanation of relationship between “HF and LF component or ratio” and vagal/sympathetic activity, there is no description about HF or LF in 11th paper.  Please correct to the appropriate citation.

  1. L309 

Number 3 should be changed to 4.

The abbreviation HRV is a non-abbreviated form, Heart Rate Variability like other chapters, as below.   4. Heart Rate Variability and Neuropsychological Functions

  1. L455

In 17th paper, the contents of the description before, “HRV has traditionally … ” does not exist.

Please correct to the appropriate citation.

We recommend that authors check for any other citation errors.

  1. L497

What does ANS body mean?  Is there any preposition missing?

Thank you very much for giving me the opportunity to review.  The characteristics of HRV shown in this paper could be used to predict health-threatening stress or mental illness as a biomarker in the near future.  Now that those biomarkers don't exist, this discussion may be a breakthrough for disease prevention and functional evaluation of stress-related illness and elucidating the mechanism of psychosomatic correlation.  The world is in such a difficult situation, but let's do our best!

Author Response

We would like to thank the Reviewer for the careful reading of the manuscript and the constructive suggestions and for allowing us to improve our paper “Is Low Heart Rate Variability associated with Emotional Dysregulation, Psychopathological Dimensions and Prefrontal Dysfunctions? An integrative view
Following, we addressed all comments and suggestions raised.

Best regards,

Anna Chiara Franquillo

Reviewer 2 Report

The article is well-written and informative. I have the following concerns:

1] In line 69, the full form of ER is not spelled out, instead it is introduced in line 108. Please introduce the full form when it first appears in the article. Similarly HRV in lines 128 and 135.

2] Line 125: “The threat activates our organism, leading it to a stress state, which is to say in a disorganization of the autonomic system's rhythmic structure and, consequently, behavioral one.” This sentence is not clear since any threat in fact activate the autonomic response in a organized manner. Please clarify with reference.

3] Line 162: “Our paper reviews the scientific works published on this matter..” Please explain what matter is referred to here.

4] The authors have not discussed the pharmacological studies. It may be better to discuss what effect the anxiolytic drugs have on emotional dysregulation and heart rate variability. This is important because the serotoninergic system along with the autonomic nervous system plays important role in emotional regulation as well as cardiopulmonary functions.

5] The figure 1 mentions frontal dysfunctions are involved in the intricate relationship. The authors may consider elaborating the frontal lobe dysfunctions that are important.

6] To make it more reader-friendly, the authors may consider incorporating tables, charts and diagrams. That will help to understand the summary of the topics discussed.

7] Is there any known effect of antiarrhythmic drugs in the relation between heart rate variability, emotion regulation, psychopathological conditions, and cognitive functions? The authors may consider discussing this aspect.

Author Response

(The authors gave the same response as above.)
